LuxS-dependent AI-2 production is not involved in global regulation of natural product biosynthesis in Photorhabdus and Xenorhabdus

Heinrich Antje K. 1
Hirschmann Merle 1
Neubacher Nick 1
Bode Helge B. h.bode@bio.uni-frankfurt.de 1 2
1 Fachbereich Biowissenschaften, Merck Stiftungsprofessur für Molekulare Biotechnologie, Goethe-Universität Frankfurt , Frankfurt am Main , Germany
2 Buchmann Institute for Molecular Life Sciences, Goethe-Universität Frankfurt , Frankfurt am Main , Germany
Keller Nancy
Electronic publication date: 2017 Jun 26
Publication date: 2017
Volume: 5
Electronic Location ID: e3471
Received 2017 Mar 21; Accepted 2017 May 26
Copyright: ©2017 Heinrich et al.
Copyright year: 2017
Copyright holder: Heinrich et al.
License: This is an open access article distributed under the terms of the Creative Commons Attribution License, which permits unrestricted use, distribution, reproduction and adaptation in any medium and for any purpose provided that it is properly attributed. For attribution, the original author(s), title, publication source (PeerJ) and either DOI or URL of the article must be cited.
License URL: https://creativecommons.org/licenses/by/4.0/

Keywords: LuxS, Autoinducer-2, Photorhabdus, Xenorhabdus, Natural products, Regulation, Secondary metabolism, Quorum sensing

Funding: European research starting grant 311477 Deutsche Forschungsgemeinschaft INST 161/810-1 Research in the Bode Laboratory is supported by a European research starting grant under grant agreement no. 311477. HBB was supported by the Deutsche Forschungsgemeinschaft for funding of an Impact II qTof mass spectrometer (INST 161/810-1). The funders had no role in study design, data collection and analysis, decision to publish, or preparation of the manuscript.

==============================
The Gram-negative bacteria Photorhabdus and Xenorhabdus are known to produce a variety of different natural products (NP). These compounds play different roles since the bacteria live in symbiosis with nematodes and are pathogenic to insect larvae in the soil. Thus, a fine tuned regulatory system controlling NP biosynthesis is indispensable. Global regulators such as Hfq, Lrp, LeuO and HexA have been shown to influence NP production of Photorhabdus and Xenorhabdus. Additionally, photopyrones as quorum sensing (QS) signals were demonstrated to be involved in the regulation of NP production in Photorhabdus. In this study, we investigated the role of another possible QS signal, autoinducer-2 (AI-2), in regulation of NP production. The AI-2 synthase (LuxS) is widely distributed within the bacterial kingdom and has a dual role as a part of the activated methyl cycle pathway, as well as being responsible for AI-2 precursor production. We deleted luxS in three different entomopathogenic bacteria and compared NP levels in the mutant strains to the wild type (WT) but observed no difference to the WT strains. Furthermore, the absence of the small regulatory RNA micA, which is encoded directly upstream of luxS, did not influence NP levels. Phenotypic differences between the P. luminescens luxS deletion mutant and an earlier described luxS deficient strain of P. luminescens suggested that two phenotypically different strains have evolved in different laboratories.

Introduction

Photorhabdus and Xenorhabdus belong to the class of entomopathogenic bacteria that are able to infect and kill insects (Goodrich-Blair & Clarke, 2007). In nature, they live in symbiosis with nematodes of the family Heterorhabditis or Steinernema, respectively, and together they infect insect larvae. As symbionts, the bacteria supply compounds that support the nematode host development, but also toxic natural products (NP) and proteins that kill the insect prey (Bode, 2009). It is easy to imagine that in the complex life style of these bacteria, inter- (bacteria-nematode, bacteria-insect, bacteria-bacteria (food predators)) and intraspecies signaling or communication plays an important role. Signal molecule dependent communication in bacteria is referred to as “quorum sensing” (QS). While in Gram-negative bacteria QS often relies on acyl homoserine lactones (AHL), summarized under the term autoinducer-1 (AI-1) (Miller & Bassler, 2001), chemically different QS molecules binding to LuxR solos, have been identified in Photorhabdus (Brachmann et al., 2013; Brameyer et al., 2015). In contrast to the QS systems of Gram-negative bacteria, Gram-positive bacteria often use modified oligopeptides as QS signals (Waters & Bassler, 2005).

With the discovery of AI-2 and its corresponding synthase, LuxS, the first possible interspecies QS system was found, as the synthase is widespread among the bacterial kingdom in Gram-positive and -negative bacteria (Pereira, Thompson & Xavier, 2013). The reason for this frequent occurrence is the enzymatic role of LuxS in the activated methyl cycle (AMC) of some bacteria in which S-adenosylhomocysteine (SAH) is recycled to recover S-adenosylmethionine (SAM) (Pereira, Thompson & Xavier, 2013). During this cycle SAH is converted to homocysteine either by a one-step reaction using the enzyme SAH hydrolase (SahH) or a two-step reaction that requires the SAH nucleosidase (Pfs) and LuxS (Winzer et al., 2002). Pfs converts SAH to S-ribosylhomocysteine (SRH), which is further transformed to homocysteine by LuxS. A “by-product” of this reaction is 4,5-dihydroxy-2,3-pentanedione (DPD), which can rearrange to R- or S-2-methyl-2,3,3,4-tetrahydroxytetrahydrofuran (R- or S-THMF), both better known as AI-2. S-THMF-borate binds to the AI-2 sensor LuxP of Vibrio harveyi, while the AI-2 receptor LsrB of Salomonella typhimurium binds the borate-free form R-THMF (Chen et al., 2002; Miller et al., 2004). Therefore, two distinct AI-2 forms are bound by two different AI-2 receptors, the first system being unique to the Vibrionaceae. The Lsr transporter (luxS regulated) is encoded by eight genes (lsrABCDFGKR), which are arranged in two operons (Taga, Miller & Bassler, 2003). As described, LsrB is the receptor for AI-2 that is then transported through the outer membrane via the membrane channel formed by LsrCD into the cell (Rezzonico, Smits & Duffy, 2012). Energy for this process is provided by the ATPase, LsrA. The kinase, LsrK, phosphorylates AI-2 in the cytoplasm and the phosphorylated AI-2 activates the transcription of the lsr operon by releasing the repressor, LsrR.

By generating luxS mutants in bacterial strains, diverse phenotypes were attributed to QS by AI-2 (Rezzonico & Duffy, 2008). With the finding that LuxS is not exclusively an AI-2 synthase, it became clear that one has to be careful when analyzing luxS mutants, not confusing metabolic effects with real QS-related phenotypes. Beside the previously mentioned LuxR solos, P. luminescens TT01 and Xenorhabdus strains also encode the AI-2 synthase LuxS in their genomes (Duchaud et al., 2003). For P. luminescens, a luxS mutant was generated and phenotypically investigated by Krin et al. (2006). Interestingly, beneath phenotypic differences in bioluminescence, oxidative stress resistance, biofilm formation, virulence and twitching motility, the luxS deficient strain showed altered carbapenem-like antibiotic production (Derzelle et al., 2002) and altered expression of a non-ribosomal peptide synthetase (NRPS) gene cluster with a yet unknown NP (Krin et al., 2006). Recently it became clear that global regulators or QS signals can alter the production of NPs in Photorhabdus/Xenorhabdus (Brameyer et al., 2014). Hfq was identified as a regulator of various NPs in P. luminescens (Tobias et al., 2017), as well as LeuO, HexA and Lrp in P. luminescens, X. nematophila and X. szentirmaii (Engel et al., 2017). In order to investigate if LuxS plays a similar role in NP regulation in these three strains, the respective luxS deletion strains were constructed and analyzed for NP production.

Material & Methods

Bacterial cultivation

Photorhabdus and Xenorhabdus strains were cultivated in LB broth (10 g/l tryptone, 5 g/l yeast extract and 5 g/l NaCl) or Schneider’s insect medium (Sigma Aldrich, St. Louis, MO, USA) with constant shaking (200 rpm unless otherwise stated) at 30 °C. All E. coli strains were grown in LB broth with shaking at 37 °C. For plate cultures LB medium contained 1.5% agar. Chloramphenicol (34 µg/ml) was added to the medium when cultivated strains carried a plasmid. During conjugation of P. luminescens and X. nematophila using E. coli S17 λpir, rifampicin (50 µg/ml) was used for selection against E. coli. When a plasmid was transferred into X. szentirmaii via conjugation ampicillin (100 µg/ml) was used for the same purpose. To enable growth of E. coli ST18, media were supplemented with 50 µg/ml δ-aminolevulinic acid (ALA). All strains used in this study are listed in Table 1.

Table 1 Bacterial strains used in this study.

Strain	Description/Genotype	Reference/source	
E. coli	
DH10B	F−araDJ39Δ(ara, leu)7697 ΔlacX74 galU galK rpsL deoRϕ8OdlacZΔM15 endAI nupG recAl mcrAΔ(mrr hsdRMS mcrBC)	Grant et al. (1990), Durfee et al. (2008)	
S17 λpir	Tp SmrrecA thi hsdRM+ RP4::2-Tc::Mu::Km Tn7, λpir phage lysogen	Simon, Priefer & Pühler (1983)	
ST18	S17 λpir ΔhemA	Thoma & Schobert (2009)	
Photorhabdus luminescens TT01	
P. luminescensG	WT, rifR (spontaneous)	Fischer-Le Saux et al. (1999), Bennett & Clarke (2005)	
P. luminescensΔluxSG	Deletion of luxS in P. luminescensG	This study	
P. luminescensΔmicAG	Deletion of micA in P. luminescensG	This study	
P. luminescensF	WT	Fischer-Le Saux et al. (1999)	
P. luminescens luxS::cmF	Deletion of luxS and insertion of a chloramphenicol resistance cassette	Derzelle et al. (2002)	
Xenorhabdus szentirmaii DSM 16338	
X. szentirmaii	WT	Gualtieri et al. (2014)	
X. szentirmaiiΔluxS	Deletion of luxS in X. szentirmaii	This study	
Xenorhabdus nematophila HGB081	
X. nematophila	WT, rifR (spontaneous)	Orchard & Goodrich-Blair (2004)	
X. nematophilaΔluxS	Deletion of luxS in X. nematophila	This study	
Enterobacter hormaechei	
ATCC 700323		ATCC®	
Enterobacter cloacae	
NEG 03 51713981	Clinical isolate		
NEG 80 51755054	Clinical isolate		
Notes.

The superscripted letters G and F differentiate between strains which were derived from the P. luminescens TT01 WT strain which is used in Germany in the Bode laboratory (G) and the P. luminescens TT01 strains which were used in France (F) (Krin et al., 2006), and were kindly provided by Evelyne Krin. All Enterobacter strains were kindly provided by Thomas A. Wichelhaus.

Table 2 Oligonucleotides used in this study.

Name	Sequence (5′ → 3′)	Purpose	
Δplu1253_up_PstI-Gib_fw	CCTCTAGAGTCGACCTGCAGTGACGA GTTTGCTAAATTGG	Amplification up- and downstream product for the deletion of luxS (plu1253) in P. luminescens	
Δplu1253_up_Gib_rev	ACTACTATGGAACAAAAAATTCAGAT TTTTCTTCAAG	
Δplu1253_do_Gib_fw	CTGAATTTTTTGTTCCATAGTAGTGAT AATATTTCGG	
Δplu1253_do_BglII-Gib_rev	TCCCGGGAGAGCTCAGATCTCCCGTA ATGAAATTGTTGG	
ΔluxS__TT01_mut_ver_fw	AGATGGAACTTGTTATCTGCC	Verification of Δplu1253	
ΔluxS__TT01_mut_ver_rev	AGTTATGCCAAAAACGATAGC	
ΔXNC1_1265_up_PstI_Gib_fw	CCTCTAGAGTCGACCTGCAGAAGCAA TTTGTAAACCGTCC	Amplification up- and downstream product for the deletion of luxS (XNC1_1265) in X. nematophila	
ΔXNC1_1265_up_Gib_rev	CTAAATACACAGATACATTACCTCCTA AGGTATCAGATT	
ΔXNC1_1265_do_Gib_fw	AGGAGGTAATGTATCTGTGTATTTAGC GGTTATCG	
ΔXNC1_1265_do_BglII-Gib_rev	TCCCGGGAGAGCTCAGATCTAATCAA ACACCAATCTATCACG	
ΔluxS__XNC1 _mut_ver_fw	TCTGTTCTTCATTCTTACGAGG	Verification of ΔXNC1_1265	
ΔluxS__XNC1 _mut_ver_rev	ATTGTTCATGCGTTGTATAGG	
ΔXSZ_luxS_up_PstI_Gib_fw	CCTCTAGAGTCGACCTGCAGCTTCAGA TGCTTTGTTACGAGG	Amplification up- and downstream product for the deletion of luxS (XSR1_140025) in X. szentirmaii	
ΔXSZ_luxS_up_Gib_rev	ATGCCAATTCCGCCATTCGTGTATGGTC	
ΔXSZ_luxS_do_Gib_fw	ACGAATGGCGGAATTGGCATTGCCTGAAG	
ΔXSZ_luxS_do_BglII-Gib_rev	TCCCGGGAGAGCTCAGATCTTTAACA CATTCTCCGCATGG	
ΔluxS__XSZ_mut_ver_fw	GACTTGCTATTTGCCTTATGC	Verification of ΔXSR1_140025	
ΔluxS__XSZ_mut_ver_rev	TCTCGAGAAAGTGACTGTCG	
ΔmicA_TT01_up_fw	TCGATCCTCTAGAGTCGACCTGCAGCA CCAATAAATCACAGAGCG	Amplification up- and downstream product for the deletion of micA region (Papamichail & Delihas, 2006) in P. luminescens	
ΔmicA_TT01_up__rev	ACAAAAAATTCAGATTCTTTTCTAGCA TCCTGTCTG	
ΔmicA_TT01_down_fw	ATGCTAGAAAAGAATCTGAATTTTTTG TGGAGATG	
ΔmicA_TT01_down_rev	GGAATTCCCGGGAGAGCTCAGATCTG TATGGTGCTTGAAGAGTTGG	
V_ ΔmicA_TT01_ii_fw	GGAAAAAATGAAGAGTCAGGG	Verification of ΔmicA	
V_ΔmicA_TT01_ii_rev	TCTGCAACACGTACTTCTGC	
V_ΔmicA_TT01_ai_fw	AGATGGAACTTGTTATCTGCC	Verification of ΔmicA	
V_ΔmicA_TT01_ai_rev	AATTAAATAAAGCCTTCAACTGG	
Notes.

_fw forward primer

_rev reverse primer

Construction of luxS deletion strains

The deletion of the LuxS encoding gene (plu1253) in P. luminescens was realized by amplifying the up- and the downstream region of this gene using primers Δplu1253_up_PstI-Gib_fw and Δplu1253_up_Gib_rev, or Δplu1253_do_Gib_fw and Δplu1253_do_BglII-Gib_rev, respectively. All oligonucleotides that were used as primers are listed in Table 2. For the upstream region, a PCR product of 919 bp was generated and the downstream PCR product had a size of 795 bp. Both PCR products were fused and integrated into the PstI and BglII linearized pCKcipB plasmid via Gibson cloning (Gibson Assembly® Master Mix, New England Biolabs). To enable Gibson cloning, primers had homologous overhangs to either the up- or the downstream product or the vector. E. coli S17 λpir cells were transformed with the Gibson assembly using electroporation. Correctness of the constructed deletion plasmid pDelta_plu1253 (Table 3) was confirmed after isolation via restriction digest and the plasmid was subsequently transferred into P. luminescens by conjugation. Conjugation of P. luminescens and chromosomal integration of the plasmid via a first homologous recombination as well as deletion of the gene of interest due to a second homologous recombination have been described previously (Brachmann et al., 2007). In order to differentiate between the desired deletion mutants and mutants genetically equal to the WT, the loss of the gene was confirmed via PCR with the primers ΔluxS__TT01_mut_ver_fw and ΔluxS__TT01_mut_ver_rev using chromosomal DNA as template. For the WT a 2,625 bp product was amplified, whereas the amplicon of the deletion mutant was only 2,096 bp long. The same strategy was used for construction of the plasmids pDelta_XNC1_1265 and pDelta_XSR1_140025 (Table 3) and the subsequent deletion of luxS in X. nematophila and X. szentirmaii. For X. nematophila, the up- and the downstream regions were amplified with the primers ΔXNC1_1265_up_PstI_Gib_fw/ΔXNC1_1265_ up_Gib_rev and ΔXNC1_1265_do_Gib_fw/ΔXNC1_1265_do_BglII-Gib_rev, yielding amplicons of 963 bp and 944 bp, respectively. The deletion of the gene was controlled with the primer pair ΔluxS__XNC1_mut_ver_fw and _rev (WT: 2626 bp and ΔluxS mutant: 2,110 bp). Upstream (857 bp) and downstream regions (793 bp) of X. szentirmaii were amplified (for this mutant only an internal 426 bp fragment of the gene was in-frame deleted) with ΔXSZ_luxS_up_PstI_Gib_fw/ΔXSZ_luxS_up_Gib_rev and ΔXSZ_luxS_do_Gib_fw/ΔXSZ_luxS_do_BglII-Gib_rev. The deletion was confirmed with primers ΔluxS__XSZ_mut_ver_fw and ΔluxS__XSZ_mut_ver_rev (WT: 2382 bp and ΔluxS mutant: 1,956 bp). Deletion of micA in P. luminescens was performed applying minor changes to the protocol described above. After Gibson cloning of the deletion plasmid, pDelta_micA, using the 884 bp upstream fragment (amplified with the primers ΔmicA_TT01_up_fw and_rev), the 840 bp downstream fragment (amplified with the primers ΔmicA_TT01_down_fw and _rev) and PstI and BglII linearized pCKcipB plasmid in one assembly reaction, the assembly mixture was used to transform E. coli ST18 cells. For cultivation of E. coli ST18 cells, 50 µg/ml ALA was added to the media. Conjugation of the plasmid from ST18 cells to P. luminescens, chromosomal integration of the plasmid, excision of the plasmid via second homologous recombination and counter selection with sucrose were performed as described above. Deletion of micA was confirmed with primers V_ΔmicA_TT01_ai_fw and _rev binding outside of the amplified region (WT: 1,879 bp and ΔmicA: 1,760 bp). Due to the small size of the deleted region, additional verification primers, binding closer to the deleted region, were used. V_ΔmicA_TT01_ii_fw and _rev leading to PCR products of 665 bp for the WT and 546 bp for ΔmicA.

Table 3 Plasmids used in this study.

Plasmid	Description	Reference/source	
pCKcipB	pDS132 (Philippe et al., 2004) based plasmid with an additional BglII restriction site, R6K ori; cmR; oriT; sacB; relaxase traI	Nollmann et al. (2015)	
pDelta_plu1253	pCKcipB based deletion plasmid encoding fused plu1253 up- (919 bp) and downstream (795 bp) regions	This study	
pDelta_XNC1_1265	pCKcipB based deletion plasmid encoding fused XNC1_1265 up- (963 bp) and downstream (944 bp) regions	This study	
pDelta_XSR1_140025	pCKcipB based deletion plasmid encoding fused XSR1_140025 up- (857 bp) and downstream (793 bp) regions	This study	
pDelta_micA	pCKcipB based deletion plasmid encoding fused micA up- (884 bp) and downstream (840 bp) regions	This study	

Bioinformatic analysis

The luxS gene and the lsr operon in P. luminescens subsp. laumondii TT01 (NC_005126.1), X. nematophila ATCC 19061 (NC_014228.1) and X. szentirmaii DSM 16338 (NZ_CBXF000000000.1) were identified by a tblastn (Basic Local Alignment Search Tool, NCBI) search. LuxS and the Lsr proteins of E. coli K-12 were used as queries (Accession numbers: LuxS: CQR82138.1, LsrKRACDBFG: CQR81040.1–CQR81047.1).

NP quantification

In order to compare NP production, analytical culture extracts were prepared. 10 ml LB medium with or without 2% of Amberlite® XAD-16 (Sigma-Aldrich, St. Louis, MO, USA) (XAD) were inoculated with a starting OD600 = 0.1 using an overnight culture. After 72 h of cultivation at 30 °C either XAD or ethyl acetate (EE) culture extracts were prepared as described before (Nollmann et al., 2015; Heinrich et al., 2016). XAD extracts of P. luminescens TT01 WT and ΔluxS were prepared after 48 h of cultivation. Briefly, XAD was separated from the supernatant and extracted with methanol (MeOH). After filtration, the crude extract was dried under reduced pressure. For HPLC-UV/MS analysis, extracts were dissolved in one culture volume of MeOH. For EE extracts 2 ml culture was extracted with an equal volume of EE. After phase separation 1 ml of the EE phase was dried under nitrogen flow and dissolved in 250 µl of MeOH. XAD extracts were prepared in quintuplicates and EE extract in quadruplicates. For this, five (XAD) or four (EE) individual cultures were inoculated with the same overnight culture and used for extraction. HPLC-UV/MS analysis was done as previously stated (Reimer et al., 2011). A total of 5 µl of each sample was separated on a C18-UHPLC column (Acquity UPLC BEH C18 1.7 lmRP 2.1 × 50 mm (Waters)) with a C18-pre-column (Acquity UPLC BEH C18 1.7 lmRP 2.1 × 5 mm (Waters)) using a H2O in acetonitrile (ACN) gradient. Both solvents were supplemented with 0.1% formic acid (FA). The gradient was either from 5–95% (ACN) in 16 min with a flow rate of 0.4 ml/min and 40 °C (XAD extracts) or from 5–95% in 22 min with 0.6 ml/min at 30 °C (EE extracts). Relative quantification of the NPs was performed as explained previously (Heinrich et al., 2016) using the software Bruker Compass DataAnalysis 4.3 for HPLC-MS data analysis and TargetAnalysis Version 1.3 for quantification of the peak area of the different compounds. The m∕z ratios which were used for generation of extracted ion chromatograms (EICs) for the quantification of the respective compounds are listed in Table 4.

Table 4 Identified and quantified compounds.

Name	Abbreviation	m∕z	Ion	Reference	
Isopropylstilbene	IPS	255.1	[M+H]+	Joyce et al. (2008)	
Anthraquinone 284	AQ-284	285.1	[M+H]+	Brachmann et al. (2007)	
Anthraquinone 270a	AQ-270a	271.1	[M+H]+	Brachmann et al. (2007)	
GameXPeptide A	GXP-A	586.4	[M+H]+	Bode et al. (2012)	
GameXPeptide B	GXP-B	600.4	[M+H]+	Bode et al. (2012)	
GameXPeptide C	GXP-C	552.4	[M+H]+	Bode et al. (2012)	
Photopyrone C	PPY-C	281.2	[M+H]+	Brachmann et al. (2013)	
Photopyrone D	PPY-D	295.2	[M+H]+	Brachmann et al. (2013)	
Photopyrone E	PPY-E	309.2	[M+H]+	Brachmann et al. (2013)	
Photopyrone F	PPY-F	323.3	[M+H]+	Brachmann et al. (2013)	
Desmethyl phurealipid A	dmPL-A	215.2	[M+H]+	Nollmann et al. (2015)	
Phurealipid A	PL-A	229.2	[M+H]+	Nollmann et al. (2015)	
Phurealipid C	PL-C	243.2	[M+H]+	Nollmann et al. (2015)	
Phurealipid B	PL-B	257.3	[M+H]+	Nollmann et al. (2015)	
Mevalagmapeptide	MVAP	334.8	[M+2H]++	Bode et al. (2012)	
Nematophin	NMT	273.2	[M+H]+	Cai et al. (2017a), Li, Chen & Webster (1997)	
Rhabdopeptide 1	RXP-1	574.4	[M+H]+	Reimer et al. (2013)	
Rhabdopeptide 2	RXP-2	588.4	[M+H]+	Reimer et al. (2013)	
Rhabdopeptide 3	RXP-3	687.5	[M+H]+	Reimer et al. (2013)	
Rhabdopeptide 4	RXP-4	701.5	[M+H]+	Reimer et al. (2013)	
Rhabdopeptide 5	RXP-5	800.6	[M+H]+	Reimer et al. (2013)	
Rhabdopeptide 6	RXP-6	814.6	[M+H]+	Reimer et al. (2013)	
Xenematide A	XMT-A	663.3	[M+H]+	Lang et al. (2008)	
Xenocoumacine I	XNC-I	466.3	[M+H]+	McInerney et al. (1991), Reimer et al. (2009)	
Xenocoumacine II	XNC-II	407.2	[M+H]+	McInerney et al. (1991), Reimer et al. (2009)	
Xenocoumacine III	XNC-III	405.2	[M+H]+	Reimer et al. (2009)	
Xenortide A	XP-A	410.3	[M+H]+	Lang et al. (2008)	
Xenortide B	XP-B	449.3	[M+H]+	Lang et al. (2008)	
Xenotetrapeptid	XTP	411.3	[M+H]+	Kegler et al. (2014)	
Szentiamide	SZT	838.4	[M+H]+	Ohlendorf et al. (2011), Nollmann et al. (2012)	
Xenofuranon A	XF-A	281.1	[M+H]+	Brachmann et al. (2006)	
Xenoamicin A	XAB-A	650.9	[M+2H]2+	Zhou et al. (2013)	
Xenoamicin B	XAB-B	657.9	[M+2H]2+	Zhou et al. (2013)	
Rhabdopeptide 771	RXP-771	772.6	[M+H]+	Cai et al. (2017b)	
Rhabdopeptide 884	RXP-884	885.6	[M+H]+	Cai et al. (2017b)	

Carbapenem production assay

The carbapenem plate assay was performed as described earlier (Derzelle et al., 2002). Agar plates with 72 h old spots of P. luminescens WTG, ΔluxSG, ΔmicAG, WTF and luxS::cmF were overlaid with swarm agar (0.6%) (sifin diagnostics gmbh) containing carbapenem sensitive Enterobacter strains (Table 1). The assay was performed for each strain in triplicates.

Bioluminescence measurements

Precultures were grown over night in 10 ml LB medium inoculated from a single colony. From a preculture 10 ml of LB medium was inoculated with an OD600 = 0.1 in triplicates and cultivated at 30 °C. At defined time points, OD600 was measured and 100 µl of each culture were transferred into the well of a microtiter plate (corning 96 flat bottom white, clear bottom polystyrol, -pure Grade™ S-, Ref: 781,670, BRANDplates®). Bioluminescence was measured with an Infinite 200 PRO reader (Tecan Trading AG, Männedorf. Switzerland) (Shaking linear duration: 4 s, shaking linear amplitude: 1 mm, top reading, mode: luminescence, attenuation: none, integration time: 1,000 ms, settle time: 0 ms). For comparability, bioluminescence was normalized by division through the OD600 measured at the same time point. GraphPad Prism 7.00 (GraphPad Software, Inc, La Jolla, CA, USA) was used for calculating P values (unpaired t-test).

Oxidative stress assay

Precultures were grown over night in 10 ml LB medium inoculated from a single colony. From a preculture, 40 ml of LB medium was inoculated with an OD600 = 0.25 and grown for ∼2 h until the culture had reached an OD600 = 0.5. 2 ml from the 40 ml culture was taken and H2O2 or paraquat was added in the tested concentrations, in triplicates. A total of 200 µl from the treated cultures were transferred into the well of a microtiter plate (Polystyrene (PS) Microtest Plate 96 Well.R, round bottom, Ref 82.1582.001, Sarstedt) and the OD600 was measured immediately in a SpectraMax 340PC384 Microplate Reader (SoftMax® Pro; Molecular Devices, Sunnyvale, CA, USA). To avoid concentration differences due to water evaporation, the outer rows of the microtiter plate were left empty. The cultures were cultivated at 30 °C in the microtiter plate and the OD600 was measured every two hours.

Biofilm assay

The ability to form biofilms on a plastic surface was monitored using a slightly modified version of the microtiter plate biofilm assay published by Merritt, Kadouri & O’Toole (2005). Cells of an overnight culture grown in LB medium were collected with centrifugation (2 min, 10,000× g, RT) and resuspended in Schneider’s insect medium adjusting to an OD600 = 0.6. For every strain, 100 µl was inoculated in six replicate wells (Polystyrene (PS) Microtest Plate 96 Well.R, round bottom, Ref 82.1582.001; Sarstedt, Nümbrecht, Germany) and incubated for 72 h at 30 °C in a humidified box. We cultivated the strains in Schneider’s insect medium, when performing the biofilm assay. The wells were washed twice with H2O and biofilms were stained with 0.1% crystal violet solution (solved in H2O) for 10 min. Unbound dye was removed and the stained biofilms were air dried. The amount of biofilm bound crystal violet serves as a measure for biofilm formation. Dye was dissolved using 30% acetic acid, with 100 µl of this solution transferred to a new microtiter plate (Polystyrene (PS) Microtest Plate 96 Well.F, flat bottom, Ref 82.1581.001; Sarstedt, Nümbrecht, Germany) for measuring the absorption at a wavelength of 570 nm using a microplate reader (Infinite 200 PRO reader; Tecan Trading AG, Männedorf. Switzerland).

Virulence assay

Precultures were diluted to an OD600 = 0.3 in 10 ml LB broth and grown at 30 °C with shaking (200 rpm) to an OD600 = 1.2–1.5. After harvesting the cells using centrifugation (10,000× g, 1 min, RT), cell pellets were resuspended in LBTween (0.1% Tween80) and each cell suspension was serially diluted to a final OD600 = 0.0002. 15 Galleria mellonella larvae per strain were injected with 5 µl of diluted cell suspension. Larvae were incubated at 30 °C and the number of living individuals was monitored every hour. In order to compare the LT50 values, data are presented in a Kaplan–Meier curve using GraphPad Prism 7.00 (GraphPad Software, Inc, La Jolla, CA, USA).

Determination of AI-2 in supernatants of P. luminescens cultures via GC-MS

The determination of AI-2 in the supernatants of P. luminescens was conducted in accordance with a protocol published previously (Thiel et al., 2009), which bases on the measurement of the precursor of AI-2, DPD. Briefly, every strain was inoculated in 10 ml of LB medium in triplicates from an overnight culture with an OD600 = 0.1 and cultivated for 24 h. LB medium without bacteria was used as a control. Cultures were centrifuged (4,000× g, 10 min, 4 °C) and 3 ml of the clear supernatant was transferred to a new 15 ml falcon and mixed with 1 ml derivatization reagent (0.1 M K2HPO4/KH2PO4 buffer, pH 7.2, supplemented with 50 mM o-phenylenediamine). After a 3 h incubation at RT (rolling), samples were extracted with 6 ml dichloromethane (DCM) and afterwards 4 ml of the organic (lower) phase were taken and dried under nitrogen flow. Dried samples were dissolved in 150 µl of DCM and 50 µl N-Methyl-(N-trimethylsilyl)-trifluoroacetamide (MSTFA) were added for derivatization. The reaction was carried out for 1 h at 60 °C. An Agilent gas chromatography (GC)-MS system with a 7890A gas chromatograph with a DB-5HT column (30 m by 250 µm by 0.1 µm) coupled to a 5975C mass spectrometer (scan range 40–300 m/z, EI ionization energy 70 eV) was used for analysis.

Two microliters of the sample were measured in split mode at a rate of 10:1. The helium flow rate was set to 1 ml/min. The inlet temperature was set to 300 °C. Analysis was performed with an initial oven temperature of 50 °C. The temperature was increased by 5 °C/min to 75 °C followed by 120 °C/min to 300 °C (hold for 5 min) and finally 120 °C/min to 50 °C (total runtime 15 min).

Results

Influence of luxS on NP production

It was shown more than a decade ago that LuxS is involved in the regulation of carbapenem production in P. luminescens (Derzelle et al., 2002) and the role of AI-2-signaling in P. luminescens was revealed by transcriptome and proteome analysis (Krin et al., 2006). In this transcriptomic analysis, an NRPS encoding gene with a yet unknown product was upregulated in the absence of LuxS in the mid-exponential phase (Krin et al., 2006) even though it was suggested that it is difficult to see transcriptional changes of genes responsible for NP biosynthesis due to their (often) low transcriptional level. In an attempt to examine if secondary metabolism in Photorhabdus and Xenorhabdus in general is regulated by AI-2 controlled QS or by another LuxS dependent mechanism, homologs of luxS were deleted in three different entomopathogenic strains: Photorhabdus luminescens, Xenorhabdus szentirmaii and Xenorhabdus nematophila. These strains were chosen since their NP production had been studied in detail previously and several compounds with their corresponding biosynthetic genes are known. A first comparison between the ΔluxS mutants and the respective WT strains did not indicate any obvious phenotypic differences concerning colony morphology and growth behavior. Only colonies of the P. luminescens ΔluxS mutant older than seven days had a darker pigmentation than the WT colonies on LB agar plates. NP levels of extracts were compared by HPLC-MS analysis. Culture extracts of all investigated strains were prepared after cultivation in LB medium supplemented with XAD. None of the ΔluxS mutants showed a significantly altered NP production compared to the WT strains (Fig. 1, for NP abbreviations see Table 4).

Figure 1 Comparison of NP production of wild type (black) and ΔluxS (white) strains of (A) P. luminescensG, (B) X. szentirmaii and (C) X. nematophila.

Production was normalized using the OD600 when XAD extracts were prepared and is given relative to the wild type production of each compound. Experiments were performed in quintuplicates. For details see ‘Material and Methods’.

Figure 2 (A) Occurrence of the lsr operon in 25 analyzed Xenorhabdus and Photorhabdus strains.

The phylogenetic tree is a trimmed version of an analysis described previously (Tobias NJ, Wolff H, Djahanschiri B, Grundmann F, Kronenwerth M, Shi Y-M, Simonyi S, Grün P, Shapiro-Ilan D, Pidot SJ, Stinear TP, Ebersberger I, Bode HB, 2017, unpublished data). LsrK: AI-2 kinase, LsrR: lsr operon transcriptional repressor, LsrA: AI-2 import ATP-binding protein, LsrC: AI-2 import system permease protein, LsrC: AI-2 import system permease protein, LsrB: AI-2 binding protein, LsrF: thiolase, LsrG: AI-2 degrading protein. (B) LuxS is required to build AI-2, so all strains investigated in detail in this study are able to generate a signal. When it comes to signal uptake only P. luminescens has all required genes for the internalization of AI-2.

Presence of the AI-2 transporter genes lsrABCDFGKR in entomopathogenic bacteria

As no difference in the amount of the produced NPs between P. luminescens, X. nematophila and X. szentirmaii and their corresponding ΔluxS strains were detected, we decided to analyze which of these strains can use AI-2 as a signal molecule. The presence or the absence of the transporter (lsr operon) for AI-2 uptake in each strain indicates, whether AI-2 can act as a QS molecule or if it is nothing but a “by-product” of the LuxS catalyzed reaction in the AMC in the respective strain. When the complete lsr operon is present in the bacterial genome, it is reported that AI-2 can function as a signal molecule by this bacterium (Rezzonico & Duffy, 2008). The genomes of the three strains were examined for the presence of all Lsr proteins present in E. coli K-12 by a tblastn search. While the genome of P. luminescens harbors the complete set of lsr genes, the genomes of X. nematophila and X. szentirmaii encode only parts of it (Fig. 2A). X. nematophila encodes only lsrKFG and the 3′ end of lsrB and X. szentirmaii encodes lsrKRFG. A wider analysis of 25 Xenorhabdus and Photorhabdus strains revealed that the pattern of the lsr operon is not directly reflected by the phylogeny (Fig. 2A). Comparing the different structures of the lsr cluster with the phylogeny indicates that loss of lsr genes was not caused by an initial deletion event in one common ancestor, but is the result of several individual losses in the affected strains. Although ten strains have lost parts of the operon, the kinase LsrK and the AI-2 degrading proteins LsrFG are encoded in all strains investigated. LsrF has a thiolase activity and LsrG acts as an isomerase (Marques et al., 2014). We concluded from these results that in our selected strains, AI-2 can play a role as a signal molecule only in P. luminescens as the two other strains lack the important channel proteins for AI-2 transport across the outer membrane (Fig. 2B) even if other uptake mechanisms exist (see ‘Discussion’). Therefore, all following investigations concentrated on P. luminescens.

Comparison of NP production of different P. luminescens WT and ΔluxS strains

Global NP production of the P. luminescens strain used in our laboratory was compared with the original set of strains described by Evelyne Krin and colleagues (2006), who kindly provided the strains. In order to differentiate between these two sets of strains, they are referred to as P. luminescens ΔluxSG (Germany) and P. luminescens luxS::cmF(France), using the same superscripted letters for the corresponding P. luminescens WTs as well. Visual inspection of colony morphology and pigmentation of liquid culture and colonies revealed that P. luminescensG and ΔluxSG have a stronger red pigmentation in comparison to the other strain pair (Fig. 3) and therefore general differences in the NP production were analyzed. Here, a slightly different protocol than explained above was used. The strains were cultivated without XAD in LB medium and the cultures were extracted with EE. XAD (which binds NPs from the culture supernatants and can therefore slightly enhance production) was not used to detect also minor regulatory changes. Since there were still no detectable changes in NP production between P. luminescensG and ΔluxSG despite the different extraction method (Fig. 3) the strains from France and Germany were compared. P. luminescensF produces noticeably lower amounts of IPS, AQs, GXPs and PPYs than P. luminescensG reaching from 4 ± 1% (AQ-270a) to 43  ± 15% (AQ-284) of WTG production. Only the phurealipids were produced in similar amounts (50 ± 14% to 95 ± 42%) by WTF. When comparing the NP production of P. luminescensF with luxS::cmF, minor changes can be seen for AQ production. The mutant strain has a slightly impaired AQ-284 production, but at the same time produces more AQ-270a. For luxS::cmF a higher amount of dmPL-A and PL-C than in the WTF was detected, but no differences in PL production were obvious when comparing P. luminescens ΔluxSG with WTG. HPLC-MS analysis of carbapenem production is very difficult due to known compound instability (Bonfiglio, Russo & Nicoletti, 2002) and therefore carbapenem production was monitored using an agar plate assay (Fig. 3B). In the presence of the antibiotic the growth of carbapenem sensitive Enterobacter strains is inhibited close to the P. luminescens colonies. With the three different Enterobacter strains used in this study no significant differences in the size of the inhibitions zones were detectable, when comparing TT01 WTG with ΔluxSG or TT01F with luxS::cmF. But both WT strains had additional to the clear inhibition zone a diffuse inhibition zone, which was not visible for the respective luxS deficient strain when overlaid with E. cloacae NEG 80 51755054. The inhibitions zones for TT01F were slightly bigger than for TT01G when overlaid with both clinically isolated E. cloacae strains.

Figure 3 Comparison of NP production and AI-2 precursor levels (DPD) of P. luminescensG and P. luminescensF.

(A) Quantification of NP levels. Production was normalized with the OD600 when EE extracts were prepared and is given relative to P. luminescensG WT production. Experiments were performed in quadruplicates. Pictures of the cultures were taken after 48 h of cultivation in LB broth at 30 °C. (B) Agar plate overlay assay for detection of carbapenem like antibiotic activity. (C) Detection of the AI-2 precursor DPD in supernatants of all investigated P. luminescens strains. In the dashed box all characteristic fragments of DPD detectable by GC-MS are shown all characteristic fragments of DPD detectable by GC-MS. Comparison of the strains is presented exemplary with the most abundant fragment. For better readability the scale of the chromatograms of WTF and luxS:: cmF were increased 6 fold. For cultivation conditions, extraction protocol, HPLC-MS measurement, quantification, overlay assay and AI-2 detection protocol see ‘Material and Methods’ section.

In order to check if deletion of luxS had an effect in all P. luminescens strains to the same extend when it comes to metabolite production, we quantified the produced amount of the AI-2 precursor DPD in the supernatant of 24 h old cultures. As expected, in supernatants of ΔluxSG and luxS::cmF no DPD was detectable, whereas in the respective WTs the precursor of AI-2 could be measured (Fig. 3C).

Comparison of the genotype of the different P. luminescens mutant strains

Since the P. luminescens strain pairs showed different levels of NPs, we compared the genotypes of ΔluxSG and luxS::cmF (Fig. 4A). The luxS gene is located between gshA encoding a γ-L-glutamyl-L-cysteine synthetase and plu1254, which encodes a protein with unknown function (Apontoweil & Berends, 1975). While P. luminescens ΔluxSG is a deletion of the entire luxS gene including 13 bp of the upstream region of the CDS, P. luminescens luxS::cmF has the luxS region replaced by a chloramphenicol resistance cassette (Derzelle et al., 2002). A detailed analysis showed that apart from the 5′ end of luxS and its complete upstream region, the last 37 bp of gshA including the stop codon are also deleted in this strain. A 200 bp fragment of the 3′ end of luxS was left in the genome. In P. luminescens and other enterobacteria the small non-coding RNA, micA, is located between gshA and luxS (Papamichail & Delihas, 2006; Vogel & Papenfort, 2006). In P. luminescens ΔluxSG micA is still intact, while in P. luminescens luxS::cmF it is deleted. To exclude that loss of micA was not responsible for the observed effects, we constructed the strain P. luminescens ΔmicAG, in which micA together with its upstream region is deleted. Figure 4B shows the sequence of micA with upstream region as described previously (Papamichail & Delihas, 2006).

Figure 4 The small regulatory RNA micA is encoded upstream of luxS and deletion of micA does not influence NP production.

(A) Comparison of the genotype of P. luminescensG∕F and the corresponding mutant strains. In the WT, the luxS gene (light blue) is located between gshA and plu1254. Present between luxS and gshA is the small regulatory RNA micA (green). cat (orange): chloramphenicol resistance cassette. (B) micA sequence of P. luminescens TT01 with upstream region (marked in green) as described previously (Papamichail & Delihas, 2006). The sequence start is marked by an arrow, −10 and −35 regions are boxed. The stop codon of gshA is marked by a star. (C) NP production of P. luminescensG and ΔmicAG. Details on cultivation, extract preparation, HPLC-MS measurements and NP quantification see ‘Material and Methods’ section.

MicA does not influence NP production in P. luminescensG

When the NP production of ΔmicAG and WTG were compared no changes in compound levels were detectable (Fig. 4C). This was also true for carbapenem production monitored via an agar plate assay (Fig. 3B).

Since the somehow contradicting results observed for P. luminescens ΔluxSG and P. luminescens luxS::cmF cannot easily be explained by the presence or absence of micA and phenotypic differences as the pigmentation between WTG and WTF were obvious, the strains were compared by a number of assays addressing bioluminescence, resistance against oxidative stress, biofilm formation and virulence.

Bioluminescence

The name Photorhabdus luminescens comes from its ability to produce light via a luciferase dependent reaction (Winson et al., 1998). In Vibrionaceae bioluminescence production is controlled by AI-2 and the LuxPQUOR system (Rezzonico, Smits & Duffy, 2012). Bioluminescence measurements with all P. luminescens strains were performed confirming the previous results as bioluminescence of WTF was two times higher than that of luxS::cmF after 7 h during the exponential growth phase (Fig. 5A) (Krin et al., 2006). Interestingly, the overall bioluminescence of WTG and the corresponding deletion strains was much lower at that time point, but here ΔluxSG also showed a two-fold lower light production than WTG. After 24 h, all strains had reached the same bioluminescence level and there was no difference between WTF and luxS::cmF. Remarkably, the bioluminescence of ΔluxSG was now two-fold higher than that of WTG. ΔmicAG behaved very similarly to the WT with respect to bioluminescence indicating that the observed effects can be attributed to the loss of luxS.

Figure 5 Phenotypic comparison of different P. luminescens strains.

(A) Measured bioluminescence normalized by the OD600 of WTG, ΔluxSG, ΔmicAG, WTF and luxS::cmF after 7, 24 and 48 h. (B) Growth curves (log scale) of all P. luminescens strains dependent from the treatment with H2O2 or paraquat. After 2 h of growth from OD600 = 0.25 reactants were added. Untreated: solid line/square, 10 mM H2O2: dotted line/triangle, 0.2 mM paraquat: dashed line/circle. (C) Biofilm formation on a polystyrene surface was compared after 72 h of cultivation via photometric quantification of biofilm bound dye upon crystal violet staining. (D) Comparison of virulence of WTG, ΔluxSG and ΔmicAG. Kaplan–Meier curve of 15 G. mellonella larvae infected per strain. The dashed line indicates the LT50 value. For details of all assays and bacterial cultivation see ‘Material and Methods’.

Oxidative stress assay

Oxidative stress assays were performed in order to test whether WTG and ΔluxSG or ΔmicAG behave differently. There are two different forms of oxidative stress that one can induce on bacteria - peroxide stress and superoxide stress (Farr & Kogoma, 1991). The oxidative defense response of each is distinct and involves different sets of proteins (Storz et al., 1990). Both pathways were tested with H2O2 used as an inducer of peroxide stress and paraquat as an inducer of superoxide stress. The exposure to 10 mM H2O2 did not show a significant effect on growth for any of the strains (Fig. 5B). This is in contrast to the previous results, where concentrations of 0.5 and 1 mM H2O2 already resulted in decreased growth of the WTF (Krin et al., 2006). When WTF and luxS::cmF were tested in the same assay conditions as our strains (Fig. 5B) we could not observe a growth defect upon addition of the higher concentration of H2O2. The addition of 0.2 mM paraquat led to a reduced growth in all strains. WTG, ΔluxSG and ΔmicAG were all influenced at the same level, so none of the deletions had an effect on paraquat sensitivity. However, luxS::cmF showed an approximately two-fold lower final optical density than WTF (0.54–0.85) compared to the untreated control.

Biofilm formation

An assay monitoring the ability to produce biofilms on polystyrene surfaces was performed. For that, biofilm bound crystal violet was solubilized and quantified by measuring its absorbance at 570 nm. No changes in biofilm formation were detected between ΔmicAG or ΔluxSG compared to WTG (Fig. 5C). The biofilm formation of P. luminescensG was around 3 times higher than that of P. luminescensF. In line with previous results (Krin et al., 2006) the luxS::cmF strain was impaired in biofilm formation and produced 3.5 times less biofilm than the corresponding WTF.

Virulence

Apart from its ability to produce a great variety of different small molecule NPs, P. luminescens is known for its capacity to kill a broad range of different insect larvae within one to two days using protein toxins (Bowen & Ensign, 1998). A virulence assay based on the infection of G. mellonella larvae with a low cell number of bacteria was performed and the LT50 values were calculated and compared. Since the LT50 values of the three strains (WTG (LT50 = 26 h), ΔluxSG (LT50 = 27 h) and ΔmicAG (LT50 = 25 h)) differed only by one hour and all strains killed infected insect larvae in less than 31 h, there are no significant changes in timing of the killing or in mortality rate of the insects (Fig. 5D).

Discussion

We aimed to examine the global role of the regulatory protein LuxS on NP production that originally was described to influence carbapenem production (Derzelle et al., 2002). However, no differences in NP production were observed showing that LuxS does not globally affect NP production levels in P. luminescensG, X. nematophila or X. szentirmaii. The investigated compounds are derived from several different biosynthetic pathways including NRPS (GXPs, RXPs, MVAP, NMT, XNCs, XTP, XABs, SZT, XP), polyketide synthases (PKS) (AQs, IPS, PPYs, XNCs), NPs derived from intermediates of fatty acid biosynthesis (PLs) or other biosynthetic pathways (XF-A) (for detailed mechanisms see references in Table 4). Altered AQ production in luxS::cmFcannot be directly linked to a LuxS dependent regulation. Both AQ-270a and AQ-284 are derived via different methylations from AQ-256, which is produced by the enzyme machinery encoded by the antA-I cluster. AQ-284 possesses one additional methyl group compared to AQ-270a (Brachmann et al., 2007). Thus, changes in the ratio of AQ-270a to AQ-284 may reflect impairments in the methylation pathway, due to the loss of LuxS in the AMC. Summing up these results we conclude that LuxS is not a global regulator of NP production in either Photorhabdus or Xenorhabdus.

A global analysis for the presence of the lsr operon, whose gene products are responsible for AI-2 uptake, was also performed. It is assumed that only when a bacterium has the complete set of lsr genes, it can use AI-2 as a real QS molecule (Rezzonico & Duffy, 2008; Brito et al., 2013). When comparing 25 Xenorhabdus and Photorhabdus strains with respect to the occurrence of the genes of the lsr operon, no pattern that follows the phylogeny was observable (Fig. 2A). This goes in line with an earlier study assigning lsr genes to be important for bacteria-nematode interaction (Gaudriault et al., 2006). There it was concluded that the lsr locus was present in a Photorhabdus/Xenorhabdus ancestor and was individually lost in different strains lost during evolution in different strains that are not living in symbiosis with H. bacteriophora. Presence of the entire operon in different Xenorhabdus strains indicates that H. bacteriophora symbiosis is not the only pressure to keep the cluster. However, the strains not harboring the complete operon still encode the kinase LsrK and the AI-2 degrading enzymes LsrFG. Investigations of different components of the Lsr transporter in S. typhimurium and E. coli revealed that deletions of lsrB (Taga, Miller & Bassler, 2003) or lsrCDB (Xavier & Bassler, 2005) still led to a slow uptake of AI-2. Only deletion of the kinase lsrK stopped the internalization of AI-2 from the supernatant completely. It was concluded that AI-2 can fulfill its function only in a phosphorylated state and that additional uptake mechanisms for AI-2 must exist. Since all analyzed strains still carry lsrK it is possible that these strains are still able to take up AI-2 via alternative receptors and then process the signal. The ribose-binding protein RbsB was shown to interact with AI-2 in Actinobacillus actinomycetemcomitans (James et al., 2006) and Haemophilus influenza (Armbruster et al., 2011). In X. szentirmaii rbsB is present in the genome, while in X. nematophila it is not. Additionally, the phosphoenolpyruvate phosphotransferase system (PTS), which is found in both Xenorhabdus strains, was shown to be involved in the initial uptake of AI-2 (Pereira et al., 2012). In summary, one can postulate that uptake of AI-2 is not the limiting factor and if there were effects on NP production, we would have seen it in P. luminescens, X. nematophila and X. szentirmaii, upon luxS deletion.

Although this study concentrated on the influence of the possible global regulator LuxS on three selected strains, we could not neglect the fact that “our” P. luminescens TT01 WT behaved differently from what was described in the literature (Krin et al., 2006). A number of phenotypic differences between P. luminescensG and P. luminescensF have been uncovered including different behavior in bioluminescence production. How bioluminescence is activated in P. luminescens is still unclear. What can be noted, is that WTG and WTF show a different time-dependent development of bioluminescence. In the transcriptome analyses Krin et al. (2006) observed that the expression of the luxCDABE genes was not altered in the luxS mutant strain and justified the change in bioluminescence by altered concentrations of spermidine in the cells. Spermidine can react with aldehydes, the substrate for the bioluminescence reaction, and quench the light development by scavenging the substrate. The role of spermidine in this process was concluded from reduced expression of several genes encoding proteins related to polyamine metabolism. Interestingly, luxS::cmF showed slightly enhanced phurealipid (PL) production (dmPL-A and PL-C). Upregulation of all PLs also could be seen in ΔluxSG in an additional analysis. PL biosynthesis also starts with fatty acid-derived aldehydes (Nollmann et al., 2015).

One biosynthesis gene cluster with reduced expression in the luxS::cmF mutant (plu4563-plu4568) was later shown to be responsible for the production of the Photorhabdus clumping factor Pcf (Brachmann et al., 2013; Brameyer et al., 2015). Therefore, it is not surprising that biofilm formation was affected in the luxS::cmF mutant (Krin et al., 2006). We were not able to observe impaired biofilm formation for the ΔluxSG mutant, although we saw the described reduction of biofilm formation for luxS::cmF in the performed microtiter plate assay (Fig. 5C). However, confusing and often contradicting results when analyzing the role of LuxS in biofilm formation have been described before (for summary see review (Hardie & Heurlier, 2008)).

In a computational approach analyzing the genome of P. luminescens, the small RNA micA, often assigned to regulation of the outer membrane protein OmpA (Udekwu et al., 2005; Rasmussen et al., 2005; Johansen et al., 2006), was predicted to be encoded upstream of luxS (Papamichail & Delihas, 2006). In Salmonella it was uncovered that impaired biofilm formation correlated with luxS deletion was indeed caused by impaired micA expression upon luxS deletion destroying the putative promoter region of micA (Kint et al., 2010). No such effect was observed for the ΔmicAG strain. The deletion of micA did not impair luxS expression, since the AI-2 precursor was measured in comparable amounts in WTG and in ΔmicAG, whereas it was not detectable in the supernatant of ΔluxSG (Fig. 3C).

Additionally, no influence of either LuxS or micA on superoxide and hydrogen peroxide stress was obvious besides the differences between the F and G strain (pairs). When compared to other studies investigating the effect of luxS mutations on the oxidative stress response, no consistent results exist that describe how AI-2 directly influences expression of genes involved in the oxidative stress defense. Studies with Streptococcus mutans (Wen & Burne, 2004) and Porphyromonas gingivalis (Yuan, Hillman & Progulske-Fox, 2005) showed that the corresponding luxS mutants had a higher tolerance towards H2O2. Contrary results were obtained for Campylobacter jejuni (He et al., 2008) and Yersinia pestis (Yu et al., 2013), where the luxS mutant strains were impaired in their resistance to oxidative stress. There exists also a study by Wilson et al. where they developed a model for Lactobacillus reuteri, which explains the altered expression (monitored with a microarray analysis) of redox stress involved genes by the metabolic role of LuxS (Wilson et al., 2012).

In a previously performed killing assay, a delay in killing larvae of the african cotton leafworm; Spodoptera littoralis (Krin et al., 2006) was observed, whereas in this study neither a difference in mortality, nor a difference in the timing of the killing of G. mellonella infected with the ΔluxS strain was obvious. Conflicting results upon luxS deletions in different species of the same genus have been reported before. Coulthurst et al. presented non-uniform phenotypes, including differences in NP production and virulence in luxS deficient mutants of two Serratia species and concluded that the regulatory effects of LuxS depend on the strain (Coulthurst, Kurz & Salmond, 2004). Barnad et al. summed up the phenotypes of luxS mutants in different Erwinia strains and differences in LuxS mediated regulation of virulence let them also conclude that LuxS seems to have different functions in different strains of Erwinia (Barnard & Salmond, 2007; Coulthurst, Lilley & Salmond, 2006; Laasik, Andresen & Mae, 2006). Furthermore, it must be kept in mind that some phenotypic assays which were described before with luxS::cmF and the ones which were performed in this study were conducted following slightly different protocols. One important variation is that Krin et al. have grown their strains in presence of 10 µM sodium borate since AI-2 is either a borated or non-borated molecule (Miller et al., 2004; Chen et al., 2002) and in order to avoid any shortage borate was added to the media. Like in S. typhimurium the lsr operon is made responsible for the transmission of AI-2 into the cells of P. luminescens. LsrB from S. typhimurium was shown to bind a non-borated version of AI-2 and even an inhibition of AI-2 mediated signaling was shown after addition of boric acid (Miller et al., 2004). Therefore, all experiments in this study were performed without addition of boric acid but using normal glassware.

Conclusion

In summary, we show that LuxS is not involved in the global regulation of NPs in P. luminescens, X. nematophila and X. szentirmaii as analyzed by HPLC-MS analysis and NP quantification. The overlay assay for the detection of carbapenem like antibiotics did not indicate any significant differences in antibiotic production of the WT and the respective mutant strains, when comparing the results seen for three different Enterobacter strains. Additionally, the known regulatory RNA micA does not influence NP production in P. luminescens.

Another result of the comparison of the different F and G strains is that these strains most likely have evolved independently in the different laboratories and therefore show very different phenotypes. Similar to our observations concerning the role of LuxS in P. luminescensG and P. luminescensF, divergent experimental outcomes for the role of Lrp in regulation of IPS biosynthesis have been explained with genomic changes in the strains used by two different groups (Kontnik, Crawford & Clardy, 2010; Lango-Scholey et al., 2013). Genome sequencing of Photorhabdus luminescens TT01 revealed that phage remnants make up already 4% of the entire genome. Additional to that, 195 IS/IS fragments and 711 ERIC (enterobacterial repetitive intergenic consensus) sequences have been found (Hulton, Higgins & Sharp, 1991; Duchaud et al., 2003). The huge amount of mobile genetic elements underscores the idea of a very flexible genome with rearrangements occurring often (Duchaud et al., 2003). Since in most laboratories (including ours) entomopathogenic bacteria are usually not grown with their host nematode, such changes might occur quickly as is also shown recently in experimental evolution experiments (Morran et al., 2016). The effects of inter-laboratory evolution were also revealed by a comparative analysis of nine laboratory “wild type” strains of the model organism Myxococcus xanthus DK1622 (Bradley et al., 2016).

We thank Evelyne Krin for kindly providing P. luminescens WT and the LuxS deficient strain (P. luminescens WTF and luxS::cmF). We are grateful to Thomas A. Wichelhaus and Denia Frank for providing the Enterobacter strains and their help with the carbapenem plate assay.

Additional Information and Declarations

Competing Interests

Author Contributions

Data Availability

The authors declare there are no competing interests.

Antje K. Heinrich and Merle Hirschmann conceived and designed the experiments, performed the experiments, analyzed the data, contributed reagents/materials/analysis tools, wrote the paper, prepared figures and/or tables.

Nick Neubacher performed the experiments, analyzed the data, contributed reagents/materials/analysis tools.

Helge B. Bode conceived and designed the experiments, analyzed the data, wrote the paper, reviewed drafts of the paper.

The following information was supplied regarding data availability:

Genomes are uploaded at Genbank. The accession numbers are as follows:

Photorhabdus asymbiotica (NC_012962.1)

Photorhabdus temperata subsp. thracensis (NZ_CP011104.1)

Photorhabdus luminescens TT01 (NC_005126.1)

Photorhabdus luminescens PB45.5 (LOIC00000000)

Xenorhabdus cabanillasii JM26 (GCA_000531755.1)

Xenorhabdus bovienii SS-2004 (NC_013892.1)

Xenorhabdus hominickii DSM 17903 (NZ_CP016176.1)

Xenorhabdus nematophila ATCC 19061 (NC_014228.1)

Xenorhabdus szentirmaii DSM 16338 (GCA_000531455.1)

Xenorhabdus mauleonii DSM 17908 (GCA_900113945.1)

Xenorhabdus bedingii DSM 4764 (MUBK00000000)

Xenorhabdus vietnamensis DSM 22392 (MUBJ00000000)

Xenorhabdus doucetiae FRM16 (GCA_000968195.1)

Xenorhabdus sp. 30TX1 (MKGR00000000)

Xenorhabdus sp. DL20 (MKGQ00000000).

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
