# Peer review of "LuxS-dependent AI-2 production is not involved in global regulation of natural product biosynthesis in Photorhabdus and Xenorhabdus"

_PeerJ, doi:10.7717/peerj.3471_

## Round 0.1 · original submission · Minor Revisions

· Academic Editor

Minor Revisions

We were able to obtain two expert reviewers and both recommended Minor Revisions. I concur with their remarks.

Reviewer 1 ·

Basic reporting

There are numerous (but very minor) syntax and grammatical mistakes with the English language used throughout this manuscript. Although these errors are not significant they could readily be corrected by a careful reading of the manuscript by a native English speaker e.g. line 63 should read "..LuxS is not exclusively an AI-2 synthase"

Experimental design

This reviewer is not convinced that adding water to wells during biofilm formation is appropriate. To avoid evaporation issues these assays should be carried out in a humidified box (e.g. food storage box containing tissue/filter paper saturated with water).

Validity of the findings

Statistics are required for Figure 5A

Additional comments

This is an interesting manuscript that compared the luxS mutant phenotype in 2 distinct lineages of P. luminescencs TTO1. This manuscript shows, contrary to an earlier study, that luxS (and therefore AI-2) is not involved in the regulation of NP production (and other phenotypes) in P. luminescens.

1. Figure 2: are there only 3 sequenced strains of Photorhabdus? What about strains of P. temperata?
2. does AI-2 accumulate in the supernatants of TTO1G and TTO1F cultures to the same level? Can Photorhabdus assimilate/degrade AI-2?
3. is it possible to determine carbapenem production using overlay assays?

Reviewer 2 ·

Basic reporting

In this article Heinrich and colleagues studied the role of the bacterial quorum sensing molecule, AI-2 and its synthase, LuxS, in the entomopathogenic bacteria Photorhabdus and Xenorhabdus. AI-2 is known to function in interspecies communication and controls collective functions of many bacteria, however, its function in entomopathogenic species was not fully understood. Focusing on natural product (NP) production, stress response and bioluminescence, the authors explore luxS-dependent regulation of two commonly used Photorhabdus luminescens laboratory strain. Although the results are generally negative, i.e. the authors did not discover a major role for luxS in these processes, I believe the data will be useful for researchers in the field. Major and minor comments have been listed below.

Experimental design

Major comments:

- Figure 1 and title: figure 1 shows that two NPs are differentially regulated in the absence of luxS in the French strain. Therefore the title might be misleading and should be adjusted accordingly.

- Figure 2B: How was the 3’ end of micA determined? Does micA terminate by a rho-independent process?

- Figure 3: Given the differences in strain construction of the two luxS mutant strains tested, it would be useful to include a complementation strain in these experiments in which luxS is expressed either from a plasmid, or better, from an alternative position in the chromosome.

Minor comments:

- Abstract, line 16: what is meant by “alternative quorum sensing”?

- Table 2: italicize gene names.

- Line 335: add “space” after “behave”.

- The results in Fig. 4B are hard to read and it is difficult to distinguish the symbols.

- Fig. 4C: change “mic” to “micA”

Validity of the findings

Except for the complementation strain requested above, I do not see any problems with the data and the interpretation.

---

## Round 0.2 · accepted · Accept

· Academic Editor

Accept

Your revisions met the spirit of the reviewer's comments and we are happy to accept your work.